# Recent Advances in CAR T-Cell Therapy for Patients with Chronic Lymphocytic Leukemia

**DOI:** 10.3390/cancers14071715

**Published:** 2022-03-28

**Authors:** Benjamin M. Heyman, Dimitrios Tzachanis, Thomas J. Kipps

**Affiliations:** 1Division of Regenerative Medicine, Department of Medicine, University of California San Diego, La Jolla, CA 92093, USA; 2Division of Blood and Marrow Transplantation, Department of Medicine, University of California San Diego, La Jolla, CA 92093, USA; dtzachanis@health.ucsd.edu; 3Center for Novel Therapeutics, Department of Medicine, University of California San Diego, La Jolla, CA 92037, USA; tkipps@health.ucsd.edu

**Keywords:** CAR T cells, chronic lymphocytic leukemia, novel approaches

## Abstract

**Simple Summary:**

Outcomes for patients with chronic lymphocytic leukemia (CLL) have significantly improved over the past decade with the introduction of targeted therapies. These medications have improved survival, with good tolerability. However, for patients in need of treatment who are refractory or intolerant to targeted therapies treatment options are limited and survival is poor. Chimeric antigen receptor T cell therapy (CAR T cell) holds great promise as a potential treatment for patients with high-risk CLL who fail conventional treatment; however, its use to date has been limited. Here we summarize the literature and treatment considerations of CAR T cell therapy for patients with CLL.

**Abstract:**

Chimeric antigen receptor T cells (CAR T cells) have resulted in dramatic treatment responses for patients with hematologic malignancies, resulting in improved survival for patients with intractable disease. The first patient treated with CD19 directed CAR T cell therapy had chronic lymphocytic leukemia (CLL) and achieved a complete remission. Subsequent clinical trials have focused largely on patients with other B-cell hematologic malignancies, owing to the fact that CAR T cell therapy for patients with CLL has met with challenges. More recent clinical trials have demonstrated CAR T cell therapy can be well tolerated and effective for patients with CLL, making it a potential treatment option for patients with this disease. In this article we review the background on CAR T cells for the treatment of patients with CLL, focusing on the unique obstacles that patients with CLL present for the development of adoptive T cell therapy, and the novel approaches currently under development to overcome these hurdles.

## 1. Introduction: Overview of CAR T Cell Therapy

Over the past decades there has been significant progress in the development of immunotherapeutic approaches for the treatment of patients with hematological malignancies. Patients with cancer, especially those with chronic lymphocytic leukemia (CLL), are defined as having relative immunodeficiency given the failure of one’s immune system to perform adequate surveillance. Thus, immunotherapeutic approaches that restore the awesome power of the immune system to eradicate malignancy are potentially curative for various cancers.

Chimeric antigen receptor T cells (CAR T cell), in the simplest form are T cells that have genetic material introduced with the goal to express a receptor that can recognize a specific antigen. The CAR design allows the gene-modified cell to have specificity for a tumor-associated antigen. The structure of current CAR T cells in clinical use includes the following components: (1) single-chain variable domain of an antibody termed, scFv, (2) a transmembrane domain, (3) a signal transduction domain of the T-cell receptor (TCR), allowing for intracellular activation, and (4) an intracellular costimulatory domain [1]. The scFv permits antigen specificity. When the CAR T cell binds a specific antigen, the T cell is activated through the signal transduction domain of the TCR, allowing for T-cell activation and expansion. First-generation CAR T cells used a CD3ζ as the signal transduction domain of the TCR. Thus, T-cell activation was solely dependent on interleukin-2 (IL-2) production. Unfortunately, clinical trials demonstrated poor efficacy of 1st generation CAR T cells, due to impaired co-stimulation and lack of adequate T-cell expansion. T cells require signaling through costimulatory receptors, such as CD28 and 4-1BB, in order to expand and produce a cytolytic response [2]. Second-generation CAR T cells were developed to contain a co-stimulatory domain, either 4-1BB or CD28. The addition of a co-stimulatory domain improved in vivo cytotoxicity, expansion, and persistence [3,4]. The choice of co-stimulatory domains impacts the expanded T-cell subset. In CAR T cells with a CD28 co-stimulatory domain, T-cell expansion is distinctive for effector T cells. While in those designed with a 4-1BB co-stimulatory domain, T cells display features of memory T cells [5,6]. Third-generation CAR T cells were designed with two co-stimulatory domains, CD28, and either 4-1BB, or OXO40 [7,8].

Currently, in clinical practice today, there are four FDA approved second generation CAR T cells products used for the treatment of patients with B-cell non-Hodgkin’s lymphoma (NHL). Axicabtagene Ciloleucel (axi-cel), consists of a single-chain variable fragment extracellular domain targeting CD19 proteins with CD3ζ signal transduction domain, and a CD28 co-stimulatory domain. Brexucabtagene autocel, was recently approved, and also consists of a single-chain variable fragment extracellular domain targeting CD19 proteins with CD3ζ signal transduction domain and a CD28 co-stimulatory domain [9]. Tisagenlecleucel (Tisa-cel) involves a murine anti-CD19 scFV; a CD8 transmembrane domain, a 4-1BB costimulatory domain, and a CD3ζ signal transduction domain [10]. Lisocabtagene maraleucel (Liso-cel) consists of a murine anti-CD19 scFV, a 4-1BB costimulatory domain; and a CD3ζ signal transduction domain. It is also derived from CD8+ and CD4+ central memory T-cell subsets in equal ratios [11].

## 2. Role of CAR T Cells for Patients with CLL

The first reported case of use of a second-generation CAR T cell product was reported in 2011 with administration of Tisa-cel in a patient with R/R CLL. The patient achieved a complete remission (CR), with ongoing remission at 10 months follow-up after treatment [12]. Since, this initial successful report, there has been significant progress in the administration of CAR T cells for patients with CLL. A summary of the important clinical trials/reports is presented in Table 1. To date patients with CLL have been particularly high risk. The majority have been refractory to either the bruton tyrosine kinase inhibitor (BTKi), ibrutinib, or the B-cell lymphoma 2 (BCL-2) inhibitor venetoclax. Approximately, 5–10% are also status-post allogeneic stem cell transplant. Clinically, patients who have received CAR T cell products harbor leukemia cells with high-risk features including TP53 mutations, complex cytogenetics, and the presence of an unmutated immunoglobulin heavy chain.

Initial efficacy of second-generation CAR T cell for patient with R/R CLL has demonstrated inferior responses and survival compared to those for patients with other lymphoid malignancies, including diffuse large B cell lymphoma (DLBCL) and B-cell acute lymphoblastic leukemia (B-ALL). Brentjens et al., treated eight patients with R/R CLL with anti-CD19 CAR T cells. Zero patients responded to treatment, with only two having stable disease [13]. Initial reports of Liso-cel in patients with R/R CLL demonstrated an ORR of 74%, with a CRR of 21%. The median progression free survival (PFS) was 8.5 months, and overall survival (OS) was not reached. Minimal residual disease (MRD) status in the bone marrow factored significantly in duration of response. Patients with undetectable MRD (uMRD) in the bone marrow significantly improved progression-free survival (PFS) compared to those with detectable MRD, not reached versus 8.5 months respectively. Other factors that impacted response included higher lymph node bulk and the number of previous treatments (>5.5 treatments) [14]. Initial reports of Tisa-cel in patients with R/R CLL were of three patients, who were heavily pre-treated, with two harboring TP53 mutations. All patients responded to treatment, with two achieving a CR. Updated analysis of the phase 1 study of 14 patients demonstrated an ORR of 57%, with a CRR of 29%. Four out of the 14 patients achieved uMRD in the bone marrow [15]. Initial reports of Axi-cel in patients with CLL, were in four patients with heavily pretreated CLL. Of the four patients who were treated, three obtained a CR with ongoing responses [16]. The ZUMA-8 study (NCT03624036), is currently ongoing to evaluate the safety and efficacy of Axi-cel in patients specifically with R/R CLL [17]. The transcend CLL 004 study evaluated Liso-cel in patients with R/R CLL. Among the 22 evaluable patients, the ORR was 82%, with a CRR of 45% (*n* = 10). At 18 months 50% of patients maintained their responses. At a median follow-up of 18 months, the median duration of response was not reached in patients who had achieved a response to Liso-cel, and median PFS was 18 months. MRD assessment demonstrated 75% had uMRD in the peripheral blood, and 65% had uMRD in the bone marrow [18].

## 3. Challenges for CAR T Cell Therapy for Patients with CLL

While there have been exciting advances in the use of CAR T cell therapy for patients with CLL, significant challenges still persist. These challenges can be divided into three categories: (1) Highly efficacious current standard of care regimens; (2) CLL patient comorbidities and immunodeficiency; and (3) Immunosubversion of CAR T cell product from patients with CLL (Figure 1).

As previously mentioned, the treatment for CLL over the past decade has undergone a paradigm changing shift from the administration of chemoimmunotherapy, now to the use of oral targeted therapies based on B cell receptor signaling and inhibition of Bcl-2. Both BTKi’s (i.e., ibrutinib) and venetoclax can provide durable remissions for patients with low toxicity related to treatment. Thus, these therapies have potentially a better therapeutic index then CAR T cells, because they can provide long-term disease control with potentially low/minimal toxicity from the treatment itself. Currently, patients with CLL who require CAR T cell therapy are refractory or intolerant to at least one if not both agents. Thus, by the time patients are candidates for CAR T cell therapy, they are likely to harbor leukemias that have acquired mutations (i.e., TP53 mutations and complex cytogenetic abnormalities) rendering them resistant to treatment, as well as potentially rapidly progressing with symptomatic disease by the time they require CAR T cell therapy. The combination of highly resistant disease with high disease burden has the potential to make CAR T cell therapy less efficacious with a greater likelihood for complications [22,23].

Many patients with CLL are elderly with significant comorbidities, with a median age of diagnosis of 70. For patients to be eligible for CAR T cell therapy, typically they will have been treated with both a BTKi and BCL-2 inhibitor, thus by the time they have progressed they may be older than 70 and potentially even in the eighth or ninth decade of life. A patient’s immune system may be impaired secondary to the underlying illness as well as from prior treatment that they have received for treatment of their leukemia. This may impair the ability to collect satisfactory cells to make the CAR T cell product, as well the function of the CAR T cell itself. Recent, analysis has demonstrated that while CAR T cell therapy has a similar rate of side effects for both young and old, it does appear there may be an increased incidence of neurotoxicity and delirium in patients who are older than 65 [24,25]. Thus, while age may have traditionally played into the role of in determining eligibility for patients who would be candidates for traditional cellular therapies such as autologous or allogeneic stem cell transplant, it by itself does not appear to impact outcome of toxicity. Thus, improving patient selection, and identifying patients who are particularly at high risk for having toxicity may be a more important determinant in eligibility. Biomarkers that have been demonstrated to be predictive of developing CRS and neurotoxicity after administration of CAR T cells are IFN-γ, IL-13, MIP1α, and IL-6 [26]. Thus, potentially developing biomarker driven treatment strategies to mitigate the risk of CRS and neurotoxicity, could help make CAR T cell therapy more widely applicable for patients with CLL especially the elderly. Prophylactic tociluzumab has been demonstrated to reduce the risk of CRS, but not neurotoxicity [27]. Thus, continued investigation and strategies are still required.

Furthermore, the lower efficacy of CAR T cell therapy in patients with CLL, appears in part secondary to the immunodeficiency and immunosubversion that occur in patients with CLL. Many standard therapies for CLL, including, chemotherapy, corticosteroids, and alemtuzumab, have a negative impact on T-cell function, which has the potential to further aggravate T-cell dysfunction in patients with CLL. Riches et al., demonstrated that T cells from patient with CLL exhibit an exhausted phenotype, characterized by increased expression of CD244, CD160, and programmed cell death protein (PD-1). CLL patients’ CD8+ T cells demonstrated the most significant functional defects in proliferation and cytotoxicity, with reduction in granzyme production and degranulation [28]. Hoffman et al., demonstrated that compared to healthy donors, CAR T cells derived from patients with CLL have significantly reduced expansion of CD4+ naïve CAR T cells, with increased expansion of PD-1+ CAR T cells. Expansion of CD4+ native T cell product was associated with improved response, with increased self-renewal. This is in part because CLL patients have reduced naïve T cells in the peripheral blood compared to healthy donors. Furthermore, malignant B-cells may also contribute to reduced expansion, by decreasing responsiveness of cytokine induced stimulation, and reducing expansion [29].

Fraietta et al., performed in depth transcriptome analysis of 41 patients with CLL who received CD-19 directed CAR T treatment. They demonstrated that patients who achieved a CR to CAR T cell therapy, possessed CAR T cells whose transcriptomes were enriched in genes reflecting that of early memory T-cells. Specifically, the CAR T cells produced high levels of STAT3-related cytokines; with serum IL-6 correlating with CAR T cell expansion and response. Patients whose CAR T cells had increased proportion of cells with the immunophenotype of CD8+CD27+PD-1-, had improved response. Patients whose CAR T cell products whose immunophenotypes were consistent with T-cell exhaustion, including increase expression of co-inhibitory receptors PD-1, TIM-3, and LAG-3, had inferior responses compared to those with reduced co-inhibitory expression [30].

Lastly, leukemia immune escape also plays a significant role in the reduced efficacy of CAR T cell for patients with CLL. Data principally from patients with both acute lymphoblastic leukemia and NHL demonstrated that patients who relapse after CAR T cell therapy undergo CD19 loss. This is thought to occur through two distinct mechanisms: either antigen escape or lineage switch. In antigen escape, after achieving a remission in response to CD19 CAR, patients relapse with a phenotypically similar disease that lacks surface expression of a CD19. Epitope loss through mutation or alternative splicing of CD19 is another common mechanism that can result in CD19 antigen loss. In lineage switch, the patient relapses with a genetically related but phenotypically different malignancy. Antigen loss appears to be more common for patients with NHL/CLL, while the role of lineage switch for patients with NHL or CLL is unclear [31,32,33,34,35].

Most studies published so far of CAR T cell therapy for patients with CLL have been conducted using autologous cells. This has the advantage that the cells are readily available, and that HLA matching is not required. However, the composition of cells in the peripheral blood can ultimately impact the expansion of the resultant CAR T cell product. Patients with CLL have variable composition of their peripheral blood mononuclear cells, in part due to the underlying disease itself and previous treatments, with patients frequently being T cell lymphopenic [36]. Low peripheral blood CD3 counts have been demonstrated to decrease the likelihood of collection of sufficient T cell to begin CAR T cell manufacturing [36]. Increased monocytes in the peripheral blood can impair T cell expansion and activation; with increased risk for T cell anergy and apoptosis [37,38]. Furthermore, leukemic cells themselves can directly inhibit T cell activation and expansion [39]. Inadvertent transduction of a leukemic cell was also demonstrated to lead to resistance to CAR T cell therapy [40]. Thus, purification is essential in order to obtain the optimal end product. The most common method of purification is the use of for magnetic adsorption using CD3/CD28 antibody coated beads during T cell activation, resulting in increased purity of the sample.

## 4. Strategies to Improve CAR T Cell Therapy for Patients with CLL

### 4.1. Combination Strategies

While CAR T cell therapy has demonstrated great promise for patients with CLL, strategies to improve response and outcomes are still required. One of the biggest challenges is the immunosubversion that patients with CLL have, due to their leukemia, limiting CAR T cell expansion and efficacy. Ibrutinib (IBR) which already has significantly improved outcomes for patients with CLL has demonstrated promising efficacy when administered in combination with CAR T cell therapy. Long et al., demonstrated that for patients with CLL treated with IBR there was a marked increase in CD4+ and CD8+ T cells. This effect was more prominent in effector/effector memory subsets. Ex vivo studies demonstrated that this may be due to diminished activation-induced cell death through interleukin-2-inducible T-cell kinase (ITK) inhibition. PD-1 and cytotoxic T-lymphocyte-associated protein 4 expression was markedly reduced in T cells by IBR. While the number of Treg cells remained unchanged, the ratio of these to conventional CD4+ T cells was reduced with ibrutinib [41]. Fraietta et al., were first to describe that three patients with CLL treated with IBR for ≥1 year had improved T-cell collection, CAR T cell expansion, and clinical response. The investigators found the IBR treatment decreased expression of co-inhibitory proteins (PD1 and CD200) of leukemia cells, without affecting CAR T cell function. In preclinical murine models the investigators found that co-administration of IBR with CAR T cell improved T cell expansion and murine OS [42,43].

Gill et al., initially demonstrated that the combination of IBR with anti-CD19 CAR T cell therapy CR rate of 43% and a bone marrow remission rate of 94% including a 78% MRD negative response by deep sequencing. CRS rates were common, but with mild severity compared to anti-CD19 CAR T cell therapy without ibrutinib [19]. Gauthier et al., recently reported a phase II clinical trial of concurrent IBR with CD19 CAR T-cell infusion for patients with CLL. Patients who had IBR started 2 weeks prior to leukapheresis and continued for ≥3 months after CAR T-cell infusion. Compared with CLL patients treated with CAR T cells without IBR, CAR T cells with concurrent ibrutinib were associated with lower CRS severity, with lower serum concentrations of CRS-associated cytokines, MCP-1, soluble IL-2Rα, and IL-6 despite comparable in vivo CAR T cell expansion. While not statistically significant, there did appear to be improvement in ORR, CRR, and MRD negativity with co-administration with IBR. The 1-year OS probabilities were 64% and 61% for the IBR and no-IBR treatment cohorts, respectively [20]. The TRANSCEND CLL 004 study (NCT02631044) is investigating the administration of Liso-cel with or without concurrent IBR. For the 22 patients who were administered Liso-cel without IBR, the ORR was 82%, with a CRR of 45%, and MRD negative rate of 65% in the BM. While for the 19 patients administered Liso-cel with IBR, the ORR was 95%, CRR 63%, and uMRD in the BM 89%. There also appeared to be a lower incidence of grade 3 CRS and grade 3 neurological toxicity for patients receiving IBR [18,21]. Thus, the administration of IBR in combination with CD19 CAR T cells for patients with CLL is a promising combination approach. Ongoing investigation into how IBR improves CAR T cell function, and reduces CRS are needed. However, it appears to be multifactorial with IBR being able to improve T-cell function by improving persistence of activated T cells, decreasing the Treg/CD4+ T-cell ratio, and diminishing the immune-suppressive properties of leukemic cells. Furthermore, ITK inhibition may promote a Th1 cellular immune response. This is supported by a recent series of seven patients with lymphoma who after initially having PD after CD19 CAR T cell therapy, went on to receive salvage IBR followed by reinfusion of CD19 CAR T cells, with all patients achieving a response, including six who achieved a CR [44].

While IBR has demonstrated promise as a combination strategy for CAR T cell therapy for patients with CLL, other combination strategies are also under investigation. As previously mentioned, CAR T cell products from patients with CLL have been found to have increased expression of co-inhibitory receptors on their surface, such as PD-1. Thus, co-administration of a checkpoint inhibitor may improve the efficacy of the CAR T cell product for patients with CLL. Furthermore, the choice of lymphodepleting chemotherapy may also impact the efficacy of checkpoint inhibitors in combination with CAR T cell administration. Preclinical models of solid tumors have demonstrated that by administering a more immunogenic lymphodepleting chemotherapy, oxaliplatin in combination with cyclophosphamide, activates tumor macrophages to express T-cell-recruiting chemokines, resulting in improved CAR-T cell infiltration, remodeling of the tumor microenvironment, and increased tumor sensitivity to anti-PD-L1 [45,46]. Current studies thus far have been limited to aggressive lymphomas but have been encouraging. An initial report in 12 patients demonstrated that the anti-PD-1 antibody pembrolizumab administered to patients who were refractory to CAR T-cell therapy was able to induce re-expansion of the CAR T-cell product and induce responses [47].

The phase 1 ZUMA-6 study (NCT02926833) is investigating Axi-cel in combination with the anti- PD-L1 antibody atezolizumab for patients with R/R DLBCL. Initial results of 12 patients demonstrated an ORR of 90%, with an encouraging CRR 60%. There were no new safety signals noted with combination therapy [48]. The phase 1/2 PLATFORM trial (NCT03310619) is investigating the combination of Liso-cel in association in combination with the anti- PD-L1 antibody durvalumab in patients with aggressive B-cell NHL. Preliminary reports of the first 11 patients, have demonstrated that 10/11 have had responses, including seven with a CR. Again, no new safety signals were seen with the combination approach [49]. Overall, these studies are promising: the addition of checkpoint blockade has the potential to improve response rate, and potentially lead to more durable responses. It should be noted that in the setting of both autologous and allogeneic stem cell transplant, the administration of checkpoint has led to increased immune related side effects. Specifically, patients who receive checkpoint inhibitors prior to autologous stem cell transplant for lymphoma are at increased risk for developing engraftment syndrome with subsequent hemodynamic instability and/or respiratory distress [50]. Similarly, for patients who undergo allogeneic stem cell transplant, prior to checkpoint inhibitor the risk is increased for graft versus host disease [51]. How the administration of checkpoint inhibitors will impact the safety of CAR T cell therapy is still unclear, but will need to be evaluated, especially as more allogeneic cell products become available clinically.

Lenalidomide is an immunomodulator that has demonstrated significant clinical activity for patients with lymphoid malignancies. Preclinical studies showed that the addition of lenalidomide to CD19 CAR T cells enhances the efficacy of CD19 CAR T cell in vivo. It results in the increased production of cytokine production and cytolytic activity in T cells. RNA sequencing revealed increased expression of genes associated with T -helper 1 response, cytokine production, and T cell activation with co-administration of lenalidomide [52,53]. A recent report of 30 patients with DLBCL and early progression after CAR T cell therapy demonstrated that patients who had received lenalidomide prior to day 15 post-CAR T-cell infusion had improved outcomes. For the 11 patients who started lenalidomide before day 15 post-CAR T-cell infusion, the ORR was 63.6% (7/11) vs. 18.8% in those who received lenalidomide after day 15. There was also a higher CRR of 36.4% vs. 10.4%, respectively. Early lenalidomide administration was associated with higher CAR T-cell expansion [53,54]. The ongoing phase 1/2 PLATFORM study is investigating the combination of Liso-cel in combination with lenalidomide for patients with R/R aggressive lymphomas.

### 4.2. Novel Targets

While CD-19 has been the predominant target of current CAR T cell products for patients with CLL, one of the primary means of evasion from CAR T cell therapy is via antigen escape. Thus, novel targets are needed to help overcome this resistance mechanism, as well potentially reducing toxicity from therapy by having improved specificity for the leukemia cells. Receptor Tyrosine Kinase Like Orphan Receptor 1 (ROR1), is an onco-embryonic antigen that is expressed during embryogenesis, and a variety of human cancers including CLL [55,56,57]. ROR1 expression on patient’s leukemia cells remains relatively stable throughout the course of their disease, and it not expressed on normal healthy differentiated tissue [56]. Thus, it is an attractive target for CAR T cell therapy, as it has the potential to improve specificity, efficacy, and reduce side effects of such a B-cell depletion and hypogammaglobulinemia related to treatment. In preclinical models of CLL, ROR1 CAR T-cells demonstrated selective toxicity for ROR1 expressing leukemia cells, with rapid clearance of leukemia cells in vivo [58]. A phase 1 clinical trial (NCT02706392) of ROR1 CAR T-cells for patients with both hematologic and solid tumors is currently underway.

Other potential targets include CD20 which is expressed on leukemia cells for patients with CLL. Phase 1/2 clinic trials for patient with R/R CD20+ B-cell lymphoma have demonstrated that CD20 directed CAR T-cells have a similar adverse event profile as CD19 directed CAR T cells [59,60]. Zhang et al., demonstrated that in 11 patients treated with a CD20 directed CAR T-cell for CD20+ B cell lymphoma, the ORR was 82%, and a CRR of 27%, with a median PFS >6 months [60]. Kappa or lambda light chain may also be another attractive target for patients with CLL, as it will allow for specificity for the leukemia cells, while avoiding complete B-cell aplasia. In a recent phase 1 trial clinical trial nine patients with R/R NHL or CLL were treated with k chain directed CAR T cells. No new safety events were noted, and a modest ORR of 33% was found for the cohort [61]. An ongoing phase 1 clinical trial (NCT04223765) of Kappa directed CAR T cells for patients with R/R NHL or CLL is underway, without any preliminary results thus far.

B cell maturation antigen (BCMA) is a cell membrane bound tumor necrosis factor receptor family member that is expressed on late-stage B lymphocytes and plasma cells. More recently, BCMA has been found to have a role in patients with CLL, specifically affecting overall prognosis. Soluble BCMA levels have been found to adversely correlate for both time to treatment failure and OS, independent of the CLL international prognostic index [62,63]. Thus, strategies that target BCMA may improve outcomes for patients with CLL. Recently, the first BCMA-directed CAR T cell therapy was approved for patients with R/R multiple myeloma (MM). In the phase 2 KarMMA trial, 127 patients with R/R MM after at four or more prior lines therapy were treated with idecabtagene vicleucel. The overall response rate for the efficacy evaluable population was 72%, and 28% of participants achieved a stringent complete response (sCR). The patients had a median time to response of 30 days. The median duration of response was 11 months for all patients who responded, and 19 months for those who achieved sCR. MRD—negative status was achieved in 26% of all patients who were treated and 79% who had a complete response or better [64]. The promising results of first BCMA-directed CAR T cells for patients with MM, may lead to the development of them for patients with CLL.

### 4.3. Novel CAR T Cell Constructs

Given the challenges that have presented in treating patient CLL with CAR T cell therapy, advances in the construction of CAR T cell products are ongoing. Bispecific CAR T cells have been developed where T cells that are transduced with a CAR that undergoes suboptimal activation upon binding to one antigen, require binding to a second antigen with subsequent activation of the chimeric costimulatory receptor in order to undergo activation. This design enhances specificity of the CAR T cell, potentially reducing toxicity. It may also decrease the risk of one of the principal mechanisms of resistance to CAR T cell therapy, CD19 antigen escape. A first in human trial phase 1 clinical trial of the bispecific anti-CD20 and anti-CD19 CAR T cell (LV20.19) for patients with R/R B-cell malignancies was recently published. Twenty-two patients with R/R B-cell NHL or CLL were treated with LV20.19 at a maximum dose of 2.5 × 10^6^ cells/kg. The ORR was 88%, with a CRR of 64%. However, for those who received at least one dose of 2.5 × 10^6^ cells/kg the ORR was 100%. For patients with CLL, two achieved a CR and one a PR at day +28. The median DOR was 10.1 months, but has not yet been reached for those who achieved a CR. The median overall survival for all patients was 20.3 months. There were no new safety signals, and grade ¾ CRS only occurred in one patient, while three had grade 3/4 neurotoxicity. A significant finding was that loss of the CD19 antigen was not seen in patients who relapsed or experienced treatment failure [65].

Armored, or 4th generation CAR T cells, have also been developed to help improve efficacy of CAR T cell therapy by protecting the T cells from the immuno-suppressive tumor microenvironment. “Armored CAR T Cells”, are engineered to express proteins, as an independent gene within the CAR vector [66]. T-cells Redirected towards Universal Cytokine Killing (TRUCK), are a specific type of armored CAR T-cell which secrete cytokines, to potentially enhance T-cell expansion and durability within the immunosuppressive tumor microenvironment, while limiting the potential side effects of systemic administration of cytokine-based therapy. There have been several cytokines assessed [67,68]. Cytokines that have been studied include IL-7, IL-12, IL-15, IL-18, IL-23 [68,69,70]. This may be a particularly attractive solution for patients with CLL, given decreased T-cell function.

Furthermore, the choice of co-stimulator domain can also affect cytokine production by CAR T cells. CD28 co-stimulation generates a more rapid and intense response then 4-1BB, but unfortunately less durable, with decreased cytotoxic activity and in vivo CAR T-cell persistence [6]. Preclinical models have demonstrated that combined CD28 and 4-1BB costimulation can lead to enhanced IL-2 secretion, cytolytic activity, and CAR T cell persistence [6]. “Third generation CAR T cells” that express both CD28 and 4-1BB costimulatory domains have now started to be evaluated in early phase clinical trials, with mixed results. An initial report by Enblad et al., of 15 patients with R/R NHL who were treated with a third generation CD-19 CAR T cell, had a modest CRR of 40% [71]. Two more recent phase 1 clinic trials of a CD-19 directed third generation CAR T cells, demonstrated similar response rates to second generation CD19 directed CAR T cells, with preliminary evidence of improved persistence of the CAR T product [72,73]. Thus, further clinical trials are needed to evaluate the efficacy and safety of third-generation CAR T cells, and if they are superior to second generation CAR T cells currently approved.

As previously mentioned, one of the biggest impediments for successful treatment of patients with CLL with CAR T cell therapy is decreased fitness and activity of CAR T cells due to immunosubversion. Thus, the use of allogeneic CAR T cells from healthy donors is an appealing option for patients with CLL, as it would avoid the use of patient-derived dysfunctional T cell. Furthermore, allogeneic CAR T cells have the potential to be an “off-the-shelf” product, which would not require delay in administration secondary to manufacturing and would increase the number of patients who potentially could receive the product. Limitations of allogeneic CAR T cells is that they can cause alloreactivity, resulting in graft-versus-host disease, as well as the potential for donor mediated rejection. Thus, strategies to minimize alloreactivity of the donor derived T cell are needed. Recent advances using CRISPR-Cas9 gene editing have the potential to facilitate specific gene editing and reduce donor alloreactivity. Genetically editing CD19-specific CAR T cells to eliminate expression of the endogenous αβ TCR can prevent a graft-versus-host response without compromising CAR-dependent effector functions [74,75]. Furthermore, directing a CD19-specific CAR specifically to the T-cell receptor α constant (TRAC) locus results in enhanced T-cell potency, with edited cells vastly outperforming conventionally generated CAR T cells in preclinical leukemia models [76] (Figure 2).

### 4.4. Allogeneic CAR T Cells

UCART19 is a universal anti-CD19 CAR T-cell product that has been generated by simultaneously knocking out TCR and CD52, with introduction of a CD19 directed CAR. Deletion of CD52 was performed, as planned therapy was to use alemtuzumab, anti-CD52, for lymphodepletion to reduce risk of UCART19 rejection [75]. A phase 1 clinical trial of UCART19 in 21 patients with R/R B-cell ALL demonstrated feasibility of allogeneic CAR T cells. The ORR was 67%, with a median duration of response of 4.1 months. PFS at 6 months was 27%. The side effect profile was similar to second generation CD19 CAR T cells in terms of grade 3/4 CRS and neurotoxicity, however there were two patient deaths [77]. ALLO-501 and ALLO-647 are similarly genetically modified anti-CD19 CAR T cell products in which the TCR alpha constant gene is disrupted. The ALPHA study is a phase 1 clinical trial of ALLO-501 or ALLO-647 in patients with R/R DLBCL or FL. Preliminary results from nine evaluable patients have demonstrated a favorable safety profile without any dose limiting toxicities or GvHD. The ORR for the cohort is 78%, with three CRs and four PRs. With a median follow up of 2.7 months, 4 pts have ongoing responses. PBCAR0191 is an allogeneic anti-CD19 CAR T-cell product generated with a single-step CAR knock-in and TCR knock-out. It includes a novel costimulatory domain (N6) that promotes cell expansion while maintaining the naïve cell phenotype. Interim data from the Phase 1/2a study of PBCAR0191 in patients with R/R NHL and B-cell ALL were recently presented and included data from 27 patients: 16 patients with R/R NHL and 11 patients with R/R B-ALL. Patients received either standard lymphodepletion with fludarabine (30 mg/m^2^/day for 3 days) plus cyclophosphamide (500 mg/m^2^/day for 3 days), or enhanced lymphodepletion fludarabine (30 mg/m^2^/day for 4 days) and cyclophosphamide (1000 mg/m^2^/day for 3 days). Patients who received enhanced lymphodepletion appeared to have improved CRR. At day 28 or later, 71% of NHL patients who received PBCAR0191 with enhanced lymphodepletion achieved a CR. While 33% of NHL patients using standard lymphodepletion achieved a CR. Peak PBCAR0191 expansion was increased 56-fold and was associated with a CR rate of 71% in the enhanced lymphodepletion group versus 33% in the standard lymphodepletion group. PBCAR0191 was also found to have an acceptable safety profile with no cases of GvHD, no cases of Grade ≥ 3 CRS, and no cases of Grade ≥ 3 neurotoxicity [78].

### 4.5. NK Cell Immunotherapy for CLL

Natural Killer Cells (NK Cells) are group 1 innate lymphoid cells that play a key role in antitumor and antiviral defense [79]. In patients with CLL, similar to T cells, there is impaired immune surveillance by NK cells with resultant decrease of NK cell cytolytic activity [80]. However, despite impaired NK cell leukemia cytolytic activity, NK cells appear to maintain their intrinsic functionality, and that impaired NK cell function may be a result of mismatched NK cell receptor-ligand pairing on NK cells and leukemia cells, as well as secretion of immunosuppressive cytokines such as TGF-B by leukemia cells [81,82]. This is supported by the fact the administration of monoclonal anti CD-19 or CD-20 antibodies are able to kill leukemia cells via NK cell dependent antibody dependent cell cytotoxicity (ADCC) [83]. Thus, autologous, or allogeneic NK cells may be an attractive cellular approach for the treatment approach for patients with CLL, as they appear to retain intrinsic functionality. This contrasts with T cells from patients with CLL which demonstrate T cell exhaustion and increased T cell terminal differentiation. Furthermore, unlike T cell, allogeneic NK cells would not have the potential to cause GvHD, and thus would not require genetic modification like allogeneic CAR T cells [84]. This could allow for the generation of adoptive NK cell from multiple sources including, peripheral blood, umbilical cord blood, hematopoietic stem cells, and induced pluripotent stem cell. Thus, adoptive NK cell immunotherapy could be a true “off-the-shelf” product, which would not require manufacturing delays as with current autologous CD19 CAR T cells [85].

One potential limitation of adoptive NK cell therapy is that a major disadvantage is their low persistence in the absence of cytokine support [86]. Preclinical models of cord-blood derived NK cells transduced with a CD19 directed CAR, and IL-15 to support their survival demonstrated potent antitumor activity against CD19+ leukemia cell lines and xenograft Raji lymphoma murine model. These results led to a phase 1 clinical trial of HLA-mismatched anti-CD19 CAR-NK cells derived from cord blood in 11 patients with R/R B-cell NHL or CLL [87]. As in the preclinical model, NK cells were transduced with a retroviral vector expressing genes that encode anti-CD19 CAR, interleukin-15, and inducible caspase 9 as a safety switch. Six patients had B-cell NHL while five had CLL. The administration of CAR NK cells was found to be safe, without the development of CRS, neurotoxicity, or GvHD. Of the 11 patients who were treated, eight (73%) had a response; of these patients, seven (four with lymphoma and three with CLL) had a CR. Responses were seen within 30 days, and CAR-NK cells persisted for at least 12 months [88].

## 5. Conclusions

The treatment for patients with CLL has changed dramatically over the past decade with introduction of the BCL-2 inhibitor venetoclax, and BTKis. This has led to a substantial improvement in survival, as well as reduced toxicity of treatment. However, for patients who progress or are intolerant to venetoclax or BTKis, options for treatment are limited, and targeted therapies often produce limited responses. CD19 CAR T cell therapy has been a paradigm changing treatment for patients with either B-cell NHL or B-ALL. Initial results for patients with CLL treated with CAR T cells, has demonstrated inferior responses, likely from a combination of patient comorbidities/immunodeficiency and immunosubversion of CAR T cell product from patients with CLL. Novel approaches to co-administer small molecules such as ibrutinib, lenalidomide, and checkpoint-inhibitors are showing early signs in improving outcomes for patients with CLL. Furthermore, novel constructs targeting different tumor antigens hold promise, as well as the introduction of allogeneic CAR T cells as a potential “off-the-shelf” approach to circumvent immunosubversion from autologous T cell products. As the field of CAR T cell therapy expands for patients with CLL, continued focus on patient factors, and CAR T cell design will need to be stressed to meet this unmet patient need. Moreover, the appropriate timing for administration of CAR T cell therapy will need to be further explored. Currently, studies to date have included patients resistant/intolerant to BTKIs, with many also being resistant to venetoclax as well. However, future studies investigating whether CAR T cell therapy may be applicable to high-risk patients, such as those who harbor 17p deletions/TP53 mutations or unmutated immunoglobulin heavy chain variable regions, earlier in there course of treatment will also need to be evaluated (Figure 3). This may allow for earlier administration of CAR T cell for patients, prior to reaching advanced age or developing co-morbidities that may preclude them from being eligible for therapy. Furthermore, as patients will not have been exposed to as many prior therapies with resultant T cell lymphopenia, the ability to collect and successfully manufacture “fit” T cells may also be improved.

## Figures and Tables

**Figure 1 cancers-14-01715-f001:**
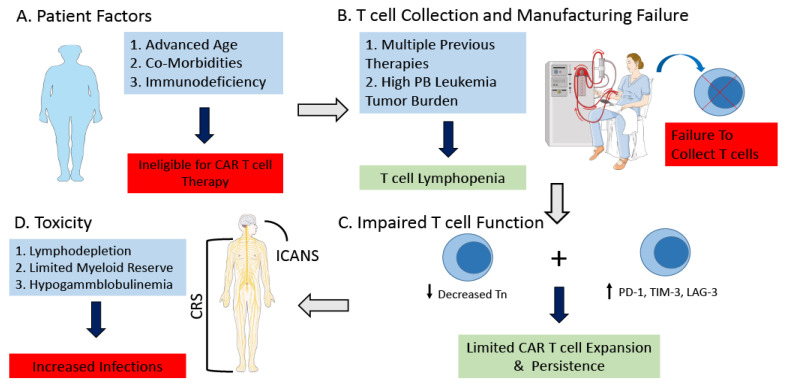
Challenges of CAR T cell Therapy for Patients with CLL. Challenges of CAR T cell therapy for patients with CLL include (**A**) Patient dependent comorbidities and immunodeficiency which can make patients ineligible for CAR T cell therapy. (**B**) Previous treatment as well as high leukemia cell burden can lead to peripheral blood T cell lymphopenia ultimately resulting in a failure to collect a satisfactory amount of T cells in order to manufacture the CAR T cell product. (**C**) Immunosubversion of the CAR T cell product from patients with CLL. T cells from patients with CLL exhibit an exhausted phenotype, characterized by increased expression of PD-1, TIM-3, and LAG-3. There is decreased naïve T cells, with significant functional defects of CD8+ T cells resulting in decreased proliferation and cytotoxicity. (**D**) Patients with CLL may be at increased risk of toxicity related to CAR T cell therapy, secondary to underlying immunosuppression, age, and co-morbidities. Abbreviations: Chronic Lymphocytic Leukemia (CLL); Peripheral Blood (PB); Naïve T cells (Tn); Programmed cell death protein 1 (PD-1); T cell immunoglobulin and mucin-domain containing-3 (TIM-3); Lymphocyte Activation Gene 3 (LAG-3); Cytokine Release Syndrome (CRS); Immune Effector cell-associated neurotoxicity syndrome (ICANS). This figure was created with images adapted from Servier Medical Art by Servier. Original images are licensed under a Creative Commons Attribution 3.0 Unported License. (smart.servier.com, accessed on 24 February 2022).

**Figure 2 cancers-14-01715-f002:**
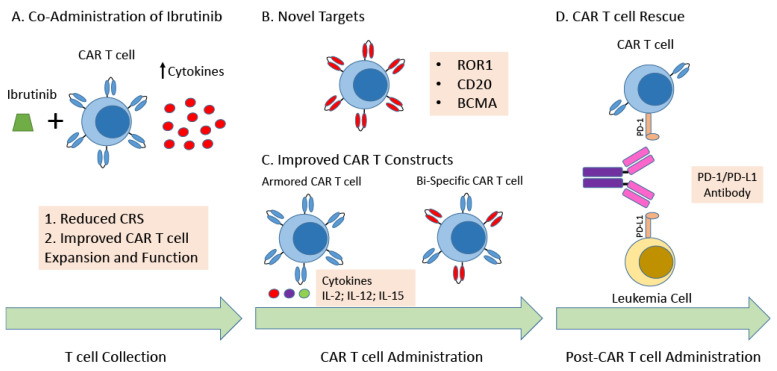
Strategies to Improve CAR T cell Therapy for Patients with CLL. (**A**) Co-administration of ibrutinib. The administration of ibrutinib prior and concomitantly with CAR T cell therapy has demonstrated improved safety with a reduction in cytokine release syndrome, and at least in preclinical models has demonstrated improved functionality of autologous CAR T cells. (**B**) Novel Targets. Targeting novel targets/antigens such as ROR1, CD20, or BCMA may be able to lead to improved specificity, resulting in improved efficacy and safety. (**C**) Improved CAR T Constructs. Current next generation CAR T cell constructs include armored CAR T cells and Bispecific CAR T cells. Armored CAR T cells are engineered to express proteins, as an independent gene within the CAR vector, usually a cytokine to improve functionality of the CAR T cell. Bispecific CAR T cells have been developed where T cells that are transduced with a CAR that undergoes suboptimal activation upon binding to one antigen, requiring binding to a second antigen with subsequent activation of the chimeric costimulatory receptor in order to undergo activation. This design enhances specificity of the CAR T cell, potentially reducing toxicity as well as immune escape. (**D**) CAR T cell Rescue. Co-administration of a checkpoint inhibitor may improve the efficacy of the CAR T cell product for patients with CLL, restoring CAR T cell expansion and functionality. Abbreviations: Chronic Lymphocytic Leukemia (CLL); Cytokine Release Syndrome (CRS); Receptor tyrosine kinase-like orphan receptor 1 (ROR1); B-cell maturation antigen (BCMA); Interleukin-2 (IL-2); Interleukin-12 (IL-12); Interleukin-15 (IL-15); Programmed cell death protein 1 (PD-1); Programmed death-ligand 1 (PD-L1).

**Figure 3 cancers-14-01715-f003:**
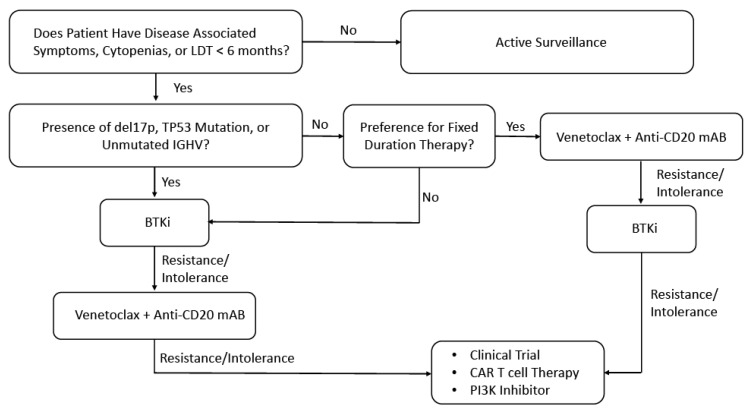
Proposed Management Algorithm for Patients with CLL. Abbreviations: Chronic Lymphocytic Leukemia (CLL); Lymphocyte doubling time (LDT); Immunoglobulin heavy-chain variable region gene (IGHV); Bruton’s tyrosine kinase inhibitor (BTKi); Monoclonal antibody (mAB); Phosphoinositide 3-kinases (PI3K).

**Table 1 cancers-14-01715-t001:** Summary of key clinical trials of CD19 directed CAR T cells for patients with CLL.

Reference	# of CLL Patients	Prior Treatment with Ibrutinib	Prior Treatment with Venetoclax	High Risk Findings	Lympho-Depletion Regimen	Combination Treatment	Efficacy	uMRD	Grade 3–5 CRS	Grade 3–5 ICANS
[12]	8	0	0	25% TP53 mutated 88% IGHV unmutated 13% Complex Karyotype	Cy	No	ORR 0% SD 38%	N/A	0%	0%
[13]	24	92%	25%	Complex Karyotype and Del 17p	Cy + Flu	No	ORR 74% CRR 21% PRR 53%	58% BM	8.3%	25%
[14]	14	N/A	N/A	43% del17p 64% IGHV unmutated	Cy + Flu	No	ORR 57% CRR 29% PRR 28%	29% BM	43%	7%
[15]	4	0	0	N/A	Cy	No	ORR 100% CRR 75% PRR 25%	N/A	25%	25%
[18]	23	100%	48%	89% (≥2) del [17p], TP53 mutation, unmutated IGHV, or complex karyotype	Cy + Flu	No	ORR 82% CRR 45% PRR 36%	75% PB 65% BM	9%	22%
[19]	19	100%	N/A	58% del17p/TP53 mutated	Cy + Flu	Yes—Ibrutinib	ORR 53% CRR 32% PRR 21%	63% BM	16%	5%
[20]	19	100%	58%	89% High Risk cytogenetics 74% del 17p 74% complex karyotype	Cy + Flu	Yes—Ibrutinib	ORR 83%	72% BM	0%	26%
[21]	19	100%	53%	73% del17p/TP53 mutated 84% IGHV unmutated	Cy + Flu	Yes—Ibrutinib	ORR 95% CRR 47% PRR 48%	89% PB 79% BM	5%	16%

Abbreviations: Undetectable Minimal Residual Disease (uMRD); Cytokine Release Syndrome (CRS); Immune Effector cell-associated neurotoxicity syndrome (ICANS); Immunoglobulin heavy-chain variable region (IGHV); Cyclophosphamide (Cy); Fludarabine (Flu); Overall Response Rate (ORR); Complete Response Rate (CRR); Partial Response Rate (PRR); Stable Disease (SD); Peripheral Blood (PB); Bone Marrow (BM).

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
