# Peer review of "Recent Advances in CAR T-Cell Therapy for Patients with Chronic Lymphocytic Leukemia"

_cancers, 2022, doi:10.3390/cancers14071715_

Round 1

Reviewer 1 Report

The figures have been changed, commented and enriched and they explain much better what they want to illustrate. The expansion of the section "Strategies to improve CAR T cell therapy for CLL" completes the reader's perspective in evaluating the results of drug combinations. However, the new paragraph at the bottom of page 6 I think is very "expanded" but it is not particularly useful in all its descriptions. It should be summarized.

Author Response

Response To Reviewer:

We appreciate the insights of the reviewer in helping to improve the manuscript. This paragraph has now been summarized. 

Reviewer 2 Report

The authors addressed all points I raised. They added new text paragraphs, redesigned graphical work and added new citations.

Overall, the manuscript has been majorly improved and is now ready for publication. 

Author Response

There were no issues to address.

This manuscript is a resubmission of an earlier submission. The following is a list of the peer review reports and author responses from that submission.

Round 1

Reviewer 1 Report

The review is well organized, clear and evaluates all the news in the specific field with adequate citations of the studies (e.g.  Brentjens, R.J. et al., Turtle, C.J. et al., Porter, D.L. et al., Flinn, I., Mancikova, V. et al, Gauthier, J. et al., Wierda, W. et al., Liu, E. et al.) The graphics of the figures could probably be improved.

The Topic is not original because it is a review but it collects all the specific state of the art on the topic with detailed updates and above all, it orders them according to a coherent and well-motivated summary path that allows you to have a well-made overview on the topic.

Compared with other published material it is a well detailed but above all well ordered and motivated overview on the specific topic.

To improve the manuscript, Probably the strategies to improve the CAR T of the CLL could be better detailed in particular with the figures that are not very beautiful and sometimes unclear. However, the main strategies in place are provided together with the ideas still in progress, so here too a good paragraph

As a review, it must collect the updated news on the subject but in particular it must give the reader an understanding of the current state of the art and future directions. It seems to me that this is done very well   There are some grammatical errors (example in line 8 of the first page "that" is repeated 2 times) but these can be easily solved by a revision of the English language.
However, they are minor issues.
My opinion is positive and I would recommend it for publication

Author Response

We are very appreciative for the helpful recommendations submitted by the reviewer.

Point 1: To improve the manuscript, Probably the strategies to improve the CAR T of the CLL could be better detailed in particular with the figures that are not very beautiful and sometimes unclear. However, the main strategies in place are provided together with the ideas still in progress, so here too a good paragraph

Response 1: We have now updated the manuscript with new figures that are improved to help assist the different aspects of the review article.  To improve the section of, “Strategies to improve CAR T cell therapy for CLL” we have added additional commentary for combination strategies, as well expanded the conclusion section to give a reader a better perspective

Point 2: There are some grammatical errors (example in line 8 of the first page "that" is repeated 2 times) but these can be easily solved by a revision of the English language.

Response 2: The manuscript has been edited to improve upon minor grammatical errors.

Once again we would like to thank the reviewer for the helpful comments.

Reviewer 2 Report

It was a pleasure reading this review on advances of CAR T cell therapy for patients with chronic lymphatic leukemia (CLL). The entire review is comprehensive and timely, it contains the contemporary literature on this subject.

Therefore, I might recommend this review for publication after clarification of some points in a revised version.

MAJOR:
1) The intrinsic problem of contaminating malignant CD19+ B-cells should be address and elaborated. MACS™ columns be mentioned as potential countermeasure.

2) Authors should speculate in a more elaborated manner on the mechanism of ibrutinib in the text, not only in Figure 2.

3) Authors should discuss the danger of check-point inhibitors to cause graft-versus-host disease (GvHD) post-allogeneic stem cell transplantation.

4) An algorithm should be drawn as figure 3 to indicate how to treat CLL patients, and what is potentially the place for CAR T cell therapy in these patients.

MINOR:

  • 42 and 42 need to be reformatted.
  • Cells and other elements must be drawn bigger with definitively larger text fonts in all figures. Authors should not depend on the French company Servier, but draw their own figures. This review should not be an advertisement for the company producing “UCARTs”, but stand for it alone.
  • In the abstract and later i.e. on page 6/6, §4, line 1, ist should read “CAR T cells” instead of “CAR T Cells”.
  • Minuscules and majuscules should be also used properly in the text inlets of both figures.
  • In the abstract, the abbreviation “CLL” should be introduced.
  • Page 2/16, line 7: two commas, one of them needs to be deleted.
  • Page 6/16, §3, line 6: “pheno-typically”, remove the hyphen.

Author Response

We are very appreciative for the helpful recommendations submitted by the reviewer.

Point 1: The intrinsic problem of contaminating malignant CD19+ B-cells should be address and elaborated. MACS™ columns be mentioned as potential countermeasure.

Response 1: We have now specifically discussed the issue of malignant contamination of CD19+ B-cells in section 3 page 6.

Point 2:  Authors should speculate in a more elaborated manner on the mechanism of ibrutinib in the text, not only in Figure 2.

Response 2: We have now expanded our discussion on the use of ibrutinib concurrently with CAR T cell therapy as well as its mechanism of action. Specifically, this can be found in section 4.1, pages 7 and 8.

Point 3:  Authors should discuss the danger of check-point inhibitors to cause graft-versus-host disease (GvHD) post-allogeneic stem cell transplantation.

Response 3: We have now addressed the danger of check-point blockade in patients receiving cell therapy under section 4.1, page 9.

Point 4:  An algorithm should be drawn as figure 3 to indicate how to treat CLL patients, and what is potentially the place for CAR T cell therapy in these patients.

Response 4: A 3rd figure with a treatment algorithm has been added.

Point 5: 42 and 42 need to be reformatted.

Response 5: The references have been reformatted.

Point 6: Cells and other elements must be drawn bigger with definitively larger text fonts in all figures. Authors should not depend on the French company Servier, but draw their own figures. This review should not be an advertisement for the company producing “UCARTs”, but stand for it alone.

Response 6: The figures have been updated and drawn.

Point 7: In the abstract and later i.e. on page 6/6, §4, line 1, ist should read “CAR T cells” instead of “CAR T Cells”. Minuscules and majuscules should be also used properly in the text inlets of both figures.In the abstract, the abbreviation “CLL” should be introduced. Page 2/16, line 7: two commas, one of them needs to be deleted. Page 6/16, §3, line 6: “pheno-typically”, remove the hyphen.

Response 7: The minor grammatical error cited above have been addressed.

We would once again thank the reviewer the very helpful comments which we believe have greatly improved the review article.